# Disgust and Self-Disgust in Eating Disorders: A Systematic Review and Meta-Analysis

**DOI:** 10.3390/nu14091728

**Published:** 2022-04-21

**Authors:** Sevgi Bektas, Johanna Louise Keeler, Lisa M. Anderson, Hiba Mutwalli, Hubertus Himmerich, Janet Treasure

**Affiliations:** 1PO59 Section of Eating Disorders, Department of Psychological Medicine, Institute of Psychiatry, Psychology & Neuroscience, King’s College London, London SE5 8AF, UK; sevgi.bektas@kcl.ac.uk (S.B.); johanna.keeler@kcl.ac.uk (J.L.K.); hiba.mutwalli@kcl.ac.uk (H.M.); janet.treasure@kcl.ac.uk (J.T.); 2Department of Psychiatry and Behavioral Sciences, University of Minnesota Medical School, 2450 Riverside Avenue South, F293, Minneapolis, MN 55454, USA; ande8936@umn.edu; 3Department of Clinical Nutrition, College of Applied Medical Sciences, Imam Abdulrahman Bin Faisal University, Dammam 34212, Saudi Arabia; 4Eating Disorders Unit, Bethlem Royal Hospital, South London and Maudsley NHS Foundation Trust (SLaM), Maudsley Hospital, Denmark Hill, London SE 8AZ, UK

**Keywords:** disgust, self-disgust, eating disorders

## Abstract

Disgust and self-disgust are aversive emotions which are often encountered in people with eating disorders. We conducted a systematic review and meta-analysis of disgust and self-disgust in people with eating disorders using Preferred Reporting Items for Systematic Reviews and Meta-Analyses (PRISMA) guidelines. The systematic review of the literature revealed 52 original research papers. There was substantial heterogeneity regarding the research question and outcomes. However, we found 5 articles on disgust elicited by food images, 10 studies on generic disgust sensitivity, and 4 studies on self-disgust, and we proceeded to a meta-analytic approach on these studies. We found that women with eating disorders have significantly higher momentary disgust feelings in response to food images (1.32; 95% CI 1.05, 1.59), higher generic disgust sensitivity (0.49; 95% CI 0.24, 0.71), and higher self-disgust (1.90; 95% CI 1.51, 2.29) compared with healthy controls. These findings indicate the potential clinical relevance of disgust and self-disgust in the treatment of eating disorders.

## 1. Introduction

Eating disorders (EDs) are complex psychiatric disorders characterized by preoccupation with weight, shape, and food [1]. The current treatment outcome for EDs is still unsatisfactory, and these disorders may persist for many years. Therefore, there is an interest in elucidating the mechanisms that underpin eating psychopathology so that more targeted and personalised treatments can be developed [2]. 

Anxiety, fear, and disgust have been thought to be involved in the problematic avoidance and/or compensatory/safety behaviours often observed in EDs [3,4,5]. Interview and self-report measures [6] have been developed to define the form of fear in eating disorders, and exposure-based treatments [7,8,9] have been designed to target these fears. However, disgust has been less extensively studied, although it is strongly linked to avoiding oral ingestion, according to Rozin and Fallon [10]. For instance, the ingestion of foods containing a potential high-pathogen load, such as rotten foods, can be a critical stimulus to elicit disgust and aversion through the evaluation of the stimulus as being contaminated. The association between disgust and a real or perceived threat from food ingestion may cause the development of unhealthy eating behaviours [11]. Furthermore, individuals may experience overwhelming feelings of disgust in response to highly palatable or calorically dense foods due to overestimating the likelihood of experiencing future weight stigma. Thus, it has been suggested that disgust may be a driving force behind food avoidance in people with AN [12].

Disgust is also experienced in relationship to socio-moral transgressions. According to Powell et al. [13], the internal world can also prompt disgust feelings through an evaluation process indicating that one’s own physical or behavioural characteristics are shameful. This can be defined as “self-disgust”. It can be associated with avoidance-based strategies towards certain parts of self that do not align with internalized social or moral norms [13]. For instance, people with physical characteristics that violate shape/weight-related socio-moral rules (i.e., thin-ideal) can experience elevated levels of self-disgust, which contribute to dietary restriction or hiding the body. Therefore, self-disgust might be a critical factor in disordered eating and body image disturbances [14]. A recent theoretical model by Glashouwer and de Jong [15] has specifically argued for the role of body-related self-disgust in AN symptomatology. 

The aim of this systematic review is to conduct a synthesis of studies that have measured aspects of disgust and self-disgust in people with EDs. 

## 2. Materials and Methods

This systematic review was carried out following Preferred Reporting Items for Systematic Reviews and Meta-Analysis 20 (PRISMA 2020) (see Appendix A) guidelines for reporting systematic reviews and meta-analyses [16]. A protocol and search strategy were prospectively registered with the Open Science Framework (OSF; available at https://osf.io/yvq3m/ Registered DOI: 10.17605/OSF.IO/YVQ3M (accessed on 27 March 2022)).

### 2.1. Eligibility Criteria for Systematic Review 

We searched for studies which included clinical samples (from psychiatric services, eating disorder services, self-help groups) of patients diagnosed with an eating disorder, in which disgust or self-disgust were also measured or were part of experiment/experimental task. All original articles published in the English language with qualitative or quantitative data included. Ineligible publication types were review articles, perspective papers, letters (without data), conference papers and book chapters, case reports, master, or doctoral theses, and animal studies. Studies classified as ineligible for inclusion were: studies focusing on obesity or metabolic disorders; studies investigating mainly normal populations or general hospital populations; and studies mainly focusing on the development, validity, or reliability of scales or diagnostic tools.

### 2.2. Eligibility Criteria for Meta-Analysis

Eligibility criteria for the systematic review described earlier were also employed for the meta-analysis. Eligible studies compared an ED group with a healthy control group. For inclusion in the meta-analysis, a minimum of four comparable studies were required. 

### 2.3. Information Sources

Studies were identified via electronic search in the following three databases: PubMed, ISI Web of Science, and APA PsycInfo (Ovid) from inception until 18 March 2022.

### 2.4. Literature Search 

Searches were conducted using combinations of the search terms: disgust or anorexia, anorexia nervosa, bulimia nervosa, binge eating disorder, ARFID, avoidant restrictive food intake disorder, avoidant/restrictive food intake disorder, pica, rumination, eating disorders AND self-disgust or anorexia, anorexia nervosa, bulimia nervosa, binge eating disorder, ARFID, avoidant restrictive food intake disorder, avoidant/restrictive food intake disorder, pica, rumination, eating disorders. Two reviewers (S.B. and H.M.) independently performed searches using the search terms. The search strategies used in each database are shown in Appendix A.

### 2.5. Study Selection and Data Extraction

The results of the literature search were first imported into Endnote, and duplicate records were removed using the web tool for systematic reviews called Rayyan [17]. Two independent reviewers (S.B. and H.M.) screened the titles and abstracts of eligible studies using Rayyan software. Disagreements over the suitability of studies for inclusion were resolved by consensus and further by discussing with other research team members (J.T., H.H., J.L.K., and L.M.A.). 

Candidate studies were reviewed as full texts and relevant data were extracted using an online Microsoft Excel spreadsheet. Extracted data included the following: title, abstract, authors, contact details, country of origin study and participant characteristics, disgust measure used (questionnaires and/or experimental tasks), and disgust outcomes with means and standard deviations. For additional data that were not reported, the first reviewer (S.B.) contacted the authors. 

## 3. Synthesis of Results

For the systematic review, studies that met the broader criteria of investigating disgust and self-disgust in individuals with EDs were reviewed, and the findings of these studies were qualitatively reported. 

For the meta-analysis, the outcome variables of interest were disgust elicited by food images, generic disgust sensitivity, and self-disgust. For the synthesis of results, Hedges’ g effect sizes were calculated for primary outcomes from the means and standard deviations. Hedges’ g results in a more precise estimate of the effect size than Cohen’s d [18], especially for studies with small sample sizes. Effect size values of 0.20 were considered “small,” 0.50 were considered “moderate,” and 0.80 were considered “large” [19]. 

### 3.1. Meta-Analyses 

Five separate meta-analyses were conducted for between-group effect sizes of: (1) disgust elicited by food images in EDs compared to controls; (2) generic disgust sensitivity in EDs compared to controls; (3) generic disgust sensitivity in AN compared to controls; (4) generic disgust sensitivity in BN compared to controls; (5) self-disgust in EDs compared to controls. Heterogeneity was suspected in all data, and so a random effects meta-analysis with the DerSimonian and Laird method [20] was used to pool Hedges’ g values. 

### 3.2. Sensitivity Analyses 

The between-study heterogeneity was assessed using Higgins I^2^ that was considered to be high when I^2^ > 75% [19]. Publication bias was assessed using Egger’s test for small study effects [21] with funnel plots. The sensitivity analyses were then conducted using the trim and fill method [22] to identify and correct funnel plot asymmetry. This method also determines if the removal of smaller studies would reduce publication bias based on re-estimated g values. Effect size calculations and analyses were computed in Stata 17 software [23].

## 4. Quality Assessment 

Two reviewers (S.B. and H.M.) independently evaluated the risk of bias for candidate studies using the Joanna Briggs Institute Critical Appraisal Tool [24]. This tool comprises several checklists for different research designs, except for experimental studies, and determines the extent to which a study has an adequate methodological quality in four categories: Yes, No, Unclear, or Not Applicable. Since one eligible study had an experimental design in this systematic review and JBI could not offer a tool for experimental studies, we made a minor modification to the checklist developed for quasi-experimental studies by adding an item that scrutinized whether the study used true randomization, taken from the checklist for Randomized Control Trial studies [24].

## 5. Results 

### 5.1. Study Selection 

Three database searches yielded 524 candidate papers, and three studies were identified through hand-searching. Following the deletion of duplicate articles, 290 articles were screened and fully read. Of these, 234 studies not meeting the eligibility criteria were dismissed. Figure 1 is a PRISMA flow diagram depicting the search for eligible studies. A total of 52 studies were eligible for inclusion in the systematic review, and 19 of these studies fitted the inclusion criteria for the meta-analyses (disgust elicited by food images *n* = 5, generic disgust sensitivity *n* = 10, self-disgust *n* = 4). All the included studies were considered to have a low risk of bias except for one study [25] that we excluded based on frequent responses to No and Unclear, showing a high risk of bias. The quality assessment of the candidate studies is shown in Appendix A.

### 5.2. Systematic Review 

The findings from the 52 individual studies included are detailed in Table 1, Table 2, Table 3, Table 4 and Table 5. The majority of studies investigated AN, BN and mixed eating disorder groups in adult females. 

#### 5.2.1. Methods to Investigate Disgust and Self-Disgust 

The eligible studies for the systematic review used various research methods which were categorized into five main methodological approaches: questionnaires or diaries to measure disgust/self-disgust, stimuli to trigger disgust, experimental tasks, brain imaging, neurophysiology, and qualitative approaches presented in Table 1, Table 2, Table 3, Table 4 and Table 5. 

##### Questionnaires

Trait disgust was measured using a range of questionnaires. Four studies applied self-report scales, such as the Differential Affect Scale [26] and the Differential Emotion Scale [27]. Two studies adapted self-report questionnaires to measure body disgust [28] in relation to pregnancy and sexuality [29]. Nine studies utilized inventories developed specifically for disgust. These inventories include the Questionnaire for Assessment of Disgust Sensitivity [30], the Questionnaire for Assessment of Disgust Proneness [31], the Disgust Propensity and Sensitivity Scale [32] or its revised form [33], the Disgust Sensitivity Questionnaire [34,35], the Disgust Scale [36,37] or its revised form [38], or the Disgust Questionnaire [39,40]. Two studies [41,42] measured a cluster of emotions of which disgust was one emotion, but they did not measure disgust specifically. The studies on self-disgust (*n* = 4) used the Multi-dimensional Self-Disgust Scale [43], the Questionnaire for Assessment of Self-Disgust [44], and the Self-Disgust Scale [33,38]. See Table 1. 

Most studies [37,45,46,47,48,49,50,51,52,53] used a visual (VAS; *n* = 10) or numeric (*n* = 1) analogue scale to rate their state disgust feelings during or after viewing/tasting stimuli or after anger induction [36]. Other studies used a Likert Scale to measure disgust ratings after taste exposure [31], Numeric Analogue Scale to measure aversive feelings (disgust and fear) in response to visual stimuli [53], or Basic Emotion Scale [36,54]. One study [55] used the DialogPad e-diary software to track emotion sequences in four categories: activation, persistence, switch, and down-regulation. See Table 2. 

##### Stimuli to Trigger Disgust

Both visual and gustatory stimuli were used. Food images included (i.e., highly palatable, or caloric foods; [31,49,50,52]) or generic disgust-eliciting food images, i.e., mouldy, or decaying foods [46,56]. Other visual stimuli included female bodies in swimming costumes in different weight categories [47,53], facial expressions of basic emotions [48,51], and generic disgust-eliciting pictures (such as animals, poor hygiene, or body products; [57]). 

Gustatory stimuli included liquid nutritional supplements [37], or solutions with a bitter taste, e.g., wormwood [31].

One study [31] measured both pre- and post-tests ratings, but other studies (*n* = 11) only reported post-test levels [37,45,46,47,48,49,50,51,52,56,57]. Uher et al. [53] rated disgust and fear together as aversive emotions; they did not provide separate findings. See Table 2. 

##### Cognitive Experimental Tasks

In total, 13 studies investigated disgust using various experimental tasks. In these tasks, the most widely used stimulus was facial expressions selected from diverse sources, including the MacBrain Face Stimulus Set [58], Pictures of Facial Affect Series by Ekman and Friesen [59,60], Matsumoto and Ekman [61], Young et al. [62], Park et al. [63] or Lundqvist et al. [64]. The “Facial Emotion Recognition Task” was the most commonly used [65,66,67,68,69,70,71].

Other experimental tasks involving facial expressions consisted of “The Visual Probe Detection Task” to investigate attentional bias [72], “The Emotional Go/No-Go Task” to yield inhibition capability [73], “The Voluntary Facial Expression Task” [74], “Facial Emotion Discrimination Task” [75], or “The Pose and Imitated Expression Task” [76]. In addition, in the study of Gagnon et al. [56], “The Temporal Bisection Task” was used to measure subjective time perception whilst showing disgusting food pictures rather than facial expressions. See Table 3.

##### Brain imaging and Neurophysiology

Task-related neural activation during the processing of various visual or gustatory stimuli among people with EDs (AN *n* = 8; BN *n* = 5; BED *n* = 1 and people with binge-eating symptoms *n* = 1) was measured in 13 studies using functional magnetic resonance imaging [49,52,53,57,77,78] and/or electrophysiological recording methodologies (*n* = 5): electroencephalography including event-related potential (ERP; [31]), evoked potential [EP; 64], late positive potentials (LPP; [47]), electromyography (EMG; [79,80]), or the simultaneous fMRI-EMG [74]. See Table 4.

##### Qualitative Studies

Four qualitative studies [54,81,82,83] evaluated how disgust was associated with eating disorder symptomatology (AN and BN *n* = 3, and EDNOS *n* = 1) following different methodologies: discourse analytic approach [82], grounded theory methodology [54,83], and theoretical framework approach [81]. Three studies collected data using semi-structured interviews [54,82,83]. One study [81] also utilised the combination of semi-structured interviews and PowerPoint slides involving the pictures of high- and low-caloric foods. See Table 5.

#### 5.2.2. Findings of Qualitative Studies

The results suggested that cognitive processes (autobiographical memories or beliefs about losing control over food; [81]), internal physical sensations, negative interpersonal experiences (i.e., judgement and criticism of others), or life events (i.e., bullying) might trigger disgust towards food or the body in people with eating disorders [54,83]. Moreover, avoidance seems to be the most relevant coping strategy [83]. For details, see Table 5.

### 5.3. Meta-Analysis 

From the studies described in the tables above, we were able to extract data suitable for a meta-analysis in three of the domains (food images, generic disgust, and self-directed disgust. 

#### 5.3.1. Disgust Elicited by Food Images 

A total of five studies with a case-control design reported disgust ratings towards food images salient for people with EDs. The five studies that were included in the meta-analysis with a total of 284 female participants, of which 139 had an ED with a mean age ranging from 16.5 to 31.4. In some studies, the diagnostic categories were separated: AN versus BN versus ED versus HC [52]; AN-R versus HC [49]; BN versus HC [50]; ED versus HC [46]; AN versus AAN versus BN versus BED [45]. State disgust was measured with a visual analogue scale (VAS). 

Figure 2 shows the meta-analysis of five studies that compared disgust elicited by food images between individuals with any ED diagnosis and HC (g = 1.32; 95% CI 1.05, 1.59; *p* < 0.001). 

#### 5.3.2. Generic Disgust Sensitivity

A total of 10 case-control studies [27,30,32,33,34,35,36,37,38,77] investigated overall disgust sensitivity in 1329 female participants, of which 767 had an ED with a mean age ranging from 21.9 to 29.7. Disgust sensitivity data are reported for the combined ED group, and two more meta-analyses were conducted on sub-samples (AN and BN groups). Eight studies [27,32,33,34,35,36,37,38] had relevant data for AN (AN *n* = 526; HC *n* = 454). One of these studies provided subgroup AN value (AN-R and AN-BP) separately, and therefore statistical data (means, standard deviations, and sample sizes) were combined for the analysis. Additionally, six studies that presented overall disgust sensitivity in people with BN relative to controls (BN *n* = 211; HC *n* = 385) were eligible meta-analysis [27,30,32,33,34,77].

Figure 3 shows the meta-analysis of 10 studies which compared generic disgust sensitivity level between individuals with any ED diagnosis and HC (g = 0.49; 95% CI 0.27, 0.71; *p* < 0.001). 

**Table 1 nutrients-14-01728-t001:** A summary of studies using questionnaires to examine disgust and self-disgust.

Author (Year) Country	Gender (*n*)	Sample Size		Age M (SD)		Study Design	Method	Main Findings	Effect Size (Cohen’s d) of Main Findings
		Clinical	Control	Clinical	Control				
**Disgust**									
**Marzola et al. (2020)** [37] ***** **Italy**	F adults	AN (33):	39	26.2 (10.3)	23.92 (2.7)	Case-control	DS	-The minimum difference between AN and HC on the baseline disgust sensitivity level.	d = −0.16
								-The minimum correlation between disgust sensitivity and eating psychopathology in AN.	d = −0.09
**Kot et al. (2021)** [38] ***** **Poland**	F adults	AN-R (29) AN-BP (34)	57	25.73 (5.99)	25.21 (5.60)	Case-control	DS-R	-The level of disgust sensitivity of AN patients was greater than HC.	d = 0.41
								-The minimum correlation between self-disgust and overall disgust sensitivity in AN.	d = −0.14
**Aharoni & Hertz (2012)** [35] ***** **Denmark**	F adults	AN-R (37) AN-BP (25)	62	General: 27.77 (6.74)		Case-control	DSQ	-The scores of overall disgust sensitivity (DS) and specific sub-scales (i.e., food, magical thinking, body products) were higher among AN than HC.	d for Overall DS = 0.76 Magical thinking = 0.84 Food = 0.94 Animal = 0.34 Body products = 0.59 Sex = 0.30 Body envelope violations = 0.11 Death = 0.13 Hygiene = 0.43
**Troop et al. (2000)** [34] ***** **UK**	F (82) M (7) adults	AN-R (16) AN-BP (12) BN (33) EDNOS (7) Binge eater (6)	15	AN-R: 21.9 (5.1) AN-BP: 29.2 (8.9) BN: 29.7 (7.5) EDNOS: 21.9 (4.2) Binge eater: 30.5 (7.8)	28.7 (7.3)	Case-control	DSQ	-The minimum difference among groups on overall disgust sensitivity.	d = 0.01
								-Drive for thinness was positively correlated with disgust sensitivity to food and magical contagion, but it was marginally associated with overall disgust sensitivity level.	d for Food = 0.77 Magical contagion = 0.52 Overall DS = 0.41
								-Bulimia symptoms were positively associated with disgust sensitivity to animals, death, body envelope violations and magical contagion, but it is marginally correlated with overall disgust sensitivity level.	d for Animal = 0.52 Death = 0.56 Body envelope violations = 0.56 Magical contagion = 0.61 Overall DS = 0.41
**Troop et al. (2002)** [40] **UK**	F adults	**Remission Group (57):** AN-R (12) AN-BP (31) BN (10) EDNOS (4)	215	General: 31.6 (10.01)	NI	Case-control	DQ	-Further analysis of a previous study by Troop et al. (2000) [34] showed that the ED group had a higher overall disgust sensitivity and disgust sensitivity to food, animal, and body products than HC.	d for Overall DS = 0.84 Food = 0.81 Animal = 0.85 Body products = 0.93
		**Clinical Group (148):** AN-R (36) AN-BP (30) BN (38) EDNOS (44)						-Both ED groups (remitted and clinical) reported a higher level of disgust towards the human body and body products and foodstuffs of animal origin than the other three domains (invertebrate animals, gastro-enteric, sexual practices).	N/A
**Davey et al. (1998)** [39] **UK**	F adolescents	AN (10)	Student girls (27)	16.1	15.2	Case-control	DQ	-The scores of patients with AN were higher in three sub-scales (foodstuffs of animal origin, human body and body products, gastro-enteric products) than HC.	d for Foodstuffs of animal origin = 1.20 Human body and body products = 0.83 Invertebrate animals = 0.02 Gastro-enteric products = 1.16 Sexual practices= 0.50
**Jiang et al. (2019)** [32] ***** **France**	F adults	AN-R (14) BN (13)	12	AN-R: 24.94 (4.67) BN: 22.50 (2.88)	24.14 (3.06)	Case-control	DPSS	-AN-R patients reported higher levels of overall disgust sensitivity than HC participants.	d = 1.20
**Bell et al. (2017)** [33] ***** **UK**	F adults	AN (270) BN (104)	217	General: 25.36 (9.67)		Case-control	DPSS-R	-The minimum correlation between self-disgust and overall disgust sensitivity for people with EDs.	d for Overall DS = 0.19
**Schienle (2003)** [30] ***** **Germany**	F (214) M (136) adults	BN (13)	150	BN: 26.0 (8.4)	40.2 (11.0)	Case-control	QADS	-The minimum difference between female BN and female HC on the disgust sensitivity level.	d = −0.15
	BN group was entirely female	Other psychiatric disorders (187)						-The minimum difference between female BN and female other psychiatric disorders on disgust sensitivity level.	N/A
**Schienle et al. (2017)** [31] **Germany**	F adults	Patients with binge-eating symptoms (36)	38	NI		Experimental	QADP	-The minimum difference between ED and HC on baseline disgust proneness levels in bitter and neutral conditions.	d for bitter condition ED vs. HC = −0.27 neutral condition ED vs. HC = −0.19
								-The minimum difference between bitter and neutral conditions for people with ED and HC.	ED bitter vs. neutral = 0.10 HC bitter vs. neutral = 0.16
**Fox and Froom (2009)** [54] **UK**	F adults	Individuals were recruited from the BEAT database (52)	N/A	31.74 (10.06)	N/A	Cross-sectional	BES **	-The positive and large correlation between ED symptomatology and state disgust.	d = 1.50
								-This association stayed large with depression and anxiety scores partialled out of the analysis.	d (depression) = 0.97 d (anxiety) = 1.05
								-After accounting for state sadness and anger within the regression model, this association disappeared.	N/A
**Fox et al. (2013)** [36] ***** **UK**	F adults	AN (22)	19	23.70 (4.20)	23.38 (3.03)	**First stage:** cross-sectional **Second stage:** case-control	**First stage:** BES ** **Second stage:** DS-R **	-State disgust was positively correlated with negative self-belief.	d = 1.61
								-State disgust was negatively correlated with positive self-belief.	d = −1.06
								The positive and large correlation between state disgust and body size/shape estimation.	d = 1.02
								-Following anger induction, the AN group reported more elevated disgust than HC.	d = 0.93
**Kollei et al. (2012)** [27] ***** **Germany**	F (105) M (25) adults	AN (32) BN (34) BDD (31)	33	AN: 26.94 (9.15) BN: 25.94 (8.25) BDD: 28.77 (8.91)	26.91 (8.48)	Case-control	DES; DES-Body	-AN and BN patients reported a higher level of overall disgust and disgust towards the body than HC.	d for AN vs. HC = 1.05 BN vs. HC = 1.0
								-The minimum difference between BDD and EDs patients (AN and BN) on emotional experiences (overall disgust and disgust towards body).	d for BDD vs. AN = 0.14 BDD vs. BN = 0.11
**Zeeck et al. (2011)** [26] **Germany**	F adults	BED and Obesity (20) Obesity without BED (23)	NW (20)	BED and Obesity: 39.3 (12.7) Obesity without BED: 45.4 (11.3)	39.7 (11.6)	Case-control	DAS	-The feeling of disgust was one of the strongest emotional experiences aggravating the association between a desire to eat and binge eating.	d = 1.74
**Bornholt et al. (2005)** [28] **Australia**	F adolescents	AN (28)	Schoolgirls (113) -Low BMI -Low–medium BMI -Medium–high BMI -High BMI	14.9 (1.8)	13.5 (1.5)	Case-control	5-point Likert Scale *** **-Item:** yuk, sick, disgust	-Adolescents with AN reported more disgust feelings about their bodies than schoolgirls with low BMI groups.	d = 0.80
								-The minimum correlation between self-concepts and disgust feelings about the body among individuals with AN.	d = 0.26
**Cooper et al. (1988)** [41] **US**	F adults	Patients with bulimia (binge–purge cycle): -with depression -without depression	N/A	25.6	N/A	Cross-sectional	The Diagnostic Survey for EDs contained a brief adjective checklist to determine emotions experienced by participants during a binge (phase 1), after a binge (phase 2) and after purging (phase 3), retrospectively.	-Factor 1, indicating feeling of guilty, disgusted, and angry, was at the highest level compared to the other three factors in the period between binge and purge. Following the purge, this level decreased by reaching the same level reported in phase 1.	d Between Phase 1 and Phase 2 for depressed people: 0.69 for non-depressed people: 0.86 Between Phase 2 and Phase 3: for depressed people: −0.86 for non-depressed people: −0.60 Between Phase 1 and Phase 3: for depressed people: −0.15 for non-depressed people: 0.23
							**-Disgust was not measured separately:** **Factor 1:** feeling of guilty, disgusted, angry **Factor 2:** energized, excited **Factor 3:** secure, relieved **Factor 4:** panicked, helpless, not calm	-The minimum difference among bulimic patients with and without depression in Factor 1 level during the binge-purge cycle.	d for difference between depressed and non-depressed people -at phase 1: 0.23 -at phase 2: −0.10 -at phase 3: 0.15
**Richson et al. (2020)** [42] **US**	F (177) M (36) Transgender (2)	AN-BP (13) BN (103) BED (14) OSFED sub-BN (73) OSFED sub-BED (16)	N/A	General: 24.73 (9.12)		Cross-sectional	EPSI-CRV used to measure Criterion B symptoms for BED	- Feeling disgusted/depressed/guilty was not a predictor of binge-eating severity.	d = 0.27
							**-Item:** feeling disgusted/depressed/guilty		
**Buvat-Herbaut et al. (1983)** [29] **France**	Young F	AN emaciation state (54) AN weight restoration (27)	Schoolgirls (288)	Range: 14–27	Range: 12–26	Case-control	Administration of the Questionnaire **Disgust-relevant Items:** -I am disgusted at being pregnant -Sexuality disgusts me	-Young females with AN reported more disgust feelings about pregnancy than HC.	N/A
								-The proportion of AN patients (at the phase of weight restoration, 28% and at the phase of emaciation, 33.3%) who were disgusted by the idea of an enlarged stomach during pregnancy was higher than that of HC (20.6%).	N/A
								-The proportion of AN patients (at the phase of weight restoration, 37.5%) who were disgusted by sexuality was higher than the proportions of AN patients in the acute phase (at the phase of emaciation, 18.4%) and HC (14%).	N/A
**Kockler et al. (2017)** [55] **Germany**	F adults	BN (20) PTSD (28) BPD (43)	28	BN: 23.70 (5.97) PTSD: 35.25 (7.53) BPD: 26.72 (7.07)	28.82 (7.47)	Case-control	DialogPad E-Diary Software measures emotion sequences in 4 categories: activation, persistence, switch, down-regulation	-BN patients experienced a most frequent change from anger to disgust than BPD, PTSD, and HC.	d for BN vs. BPD = 0.42 BN vs. PTSD = 0.47 BN vs. HC = 0.52
								-The most frequent switch was from disgust to an unspecific emotion for patients with BN relative to those with BPD and HC.	d for BN vs. BPD = 0.46 BN vs. HC = 0.56
**Self-Disgust**									
**Bell et al. (2017)** [33] ***** **UK**	F adults	AN (270) BN (104)	217	General: 25.36 (9.67)		Case-control	SDS	-ED group reported a higher level of self-disgust than HC.	d = 1.19 d for Anxiety = 0.36 Low registration = 0.35 Sensation seeking = −0.23
								-Self-disgust was positively associated with anxiety symptoms, low registration and negatively correlated with sensation seeking among the AN group.	
								-Self-disgust was positively associated with anxiety symptoms, sensation avoidance, and sensation seeking among the BN group.	d for Anxiety = 0.30 Low registration = 0.37 Sensation seeking = −0.26
**Ille et al. (2014)** [44] ***** **Austria**	F (93) M (19) adults ED group is entirely female	Clinical sample consisted of AN (16) and BN (24)	112	No ED-specific age details	31.10 (13.0)	Case-control	QASD	-Individuals diagnosed with EDs reported higher personal and behavioural disgust than HC.	d for personal disgust = 1.68 behavioural disgust = 1.59
								-For EDs patients, whereas interpersonal sensitivity, depression, and obsession were predictors for personal disgust (corrected R2 = 0.70), the best predictor of behavioural disgust was anxiety (corrected R2 = 0.26).	N/A
**Kot et al. (2021)** [38] ***** **Poland**	F adults	AN-R (29) AN-BP (34)	57	25.73 (5.99)	25.21 (5.60)	Case-control	SDS Female patients with AN aged between 18 and 45 years	-The level of self-disgust among patients with AN was greater than in HC.	d for overall self-disgust = 0.41
								-The minimum correlation between self-disgust and overall disgust sensitivity in AN and HC.	d for AN = −0.14 HC = 0.26
								-Self-disgust predicted the severity of EDs characteristics.	N/A
								-Self-disgust mediated the associations between ED characteristics and depressive symptoms and trait anxiety in AN and HC.	N/A
**Marques et al. (2021)** [43] ***** **Portugal**	F adults	62	119	32.16 (13.19)	22.45 (3.50)	Case-control	MSDS	-In comparison with the community sample, ED patients reported higher levels of self-disgust.	d = 1.71
								-Self-disgust was positively correlated with a drive for thinness and external shame and negatively correlated with self-compassion level.	d for Drive for thinness = 0.77 External shame = 1.25 Self-compassion = −1.25
								-Self-compassion played a moderator role (*b* = − 0.24; *p* = 0.033) in the relationship between self-disgust and ED symptomatology.	N/A

**Abbreviations:** NI: No Information, N/A: Not Applicable, F: Female, BN: Bulimia Nervosa, AN: Anorexia Nervosa, AN-R: Anorexia Nervosa-Restrictive Type, AN-BP: Anorexia Nervosa: Binge–Purge Type, BED: Binge Eating Disorder, EDNOS: Eating Disorder Not Otherwise Specified, OSFED: Other Specified Feeding and Eating Disorders, BDD: Body Dysmorphic Disorder, BPD: Borderline Personality Disorder, PTSD: Post-Traumatic Stress Disorder, ED: Eating Disorder, EDs: Eating Disorders, HC: Healthy Control, BMI: Body Mass Index, Age M: Age Mean, SD: Standard Deviation, DS: Disgust Scale, DS-R: Disgust Scale-Revised, DSQ: Disgust Sensitivity Questionnaire, DQ: Disgust Questionnaire, DPSS: Disgust Propensity and Sensitivity, DPSS-R: Disgust Propensity and Sensitivity-Revised, QADP: Questionnaire for the Assessment of Disgust Propensity, QADS: Questionnaire for the Assessment of Disgust Sensitivity, BES: Basic Emotion Scale, DAS: Differential Affect Scale, DES: Differential Emotion Scale, EPSI-CRV: Eating Pathology Symptoms Inventory-Clinician Rated Version, SDS: Self-Disgust Scale, QASD: Questionnaire for the Assessment of Self-Disgust, MSDS: The Multi-dimensional Self-Disgust Scale. **Footnotes:** * The study included in our meta-analyses. ** The scale was used to measure state disgust. *** It was not clear whether the scale measured state or trait disgust.

**Table 2 nutrients-14-01728-t002:** A summary of studies using stimuli to trigger disgust.

Author (Year) Country	Gender (*n*)	Sample Size		Age M (SD)		Study Design	Method	Main Findings	Effect Size (Cohen’s d) of Main Findings
		Clinical	Control	Clinical	Control				
**Uher et al. (2005)** [53] ***** **UK**	F adults	BN (9) AN (13)	18	BN: 29.6 (9.3) AN: 25.4 (10.2)	26.6 (8.6)	Case-control	Numeric Analogue Scale of 1–7 for disgust and fear to **visual stimuli: underweight, normal, and overweight female bodies in swimming costumes**	-Each body shape category was more aversive to ED patients than it was to HC. This effect was more marked in AN than in BN.	d for Underweight = 0.99 Normal weight = 1.30 Overweight = 2.13
								-The most aversive body shape category was underweight for HC, while it was overweight for ED group.	N/A
								-AN patients reported more aversion to normal-weight bodies compared to BN and HC.	N/A
**Schienle et al. (2017)** [31] **Germany**	F adults	Patients with binge-eating symptoms (36)	38	NI		Experimental	9-point Likert Scale	-Aftertaste ratings showed wormwood was perceived as more disgusting than water by each participant.	d = −6.32
							**visual stimuli (food pictures) and gustatory fluid stimuli, including water (neutral tastant) and wormwood (bitter/aversive tastant)**		
**Marzola et al. (2020)** [37] ****** **Italy**	F adults	AN-R (21) AN-BP (12)	39	26.2 (10.3)	23.92 (2.7)	Case-control	VAS **(disgust towards gustatory stimuli: supplement)**	-The supplement induced more food-related disgust than the juice in patients with AN in comparison with HC.	d = 0.51
**Joos et al. (2012)** [51] **Germany**	F adults	AN (23) BN (29) Depression (35)	25	AN: 24.0 (4.7) BN: 26.2 (6.3) Depression: 27.6 (5.7)	27.4 (5.5)	Case-control	VAS **(disgust towards visual stimuli: clear or blended emotional facial expressions)**	-Medium difference among groups (EDs vs. HC; EDs vs. Depression) on disgust responses to angry facial stimuli.	d for EDs vs. HC = 0.40 ED vs. Depression = 0.38
**Joos et al. (2009)** [48] ****** **Germany**	F adults	AN-R (15) BN (19)	25	AN-R: 25.0 (4.5) BN: 25.4 (6.4)	27.4 (5.5)	Case-control	VAS **(disgust towards visual stimuli: facial expressions of different clear or blended emotions)**	-The small difference between AN-R and BN on disgust response towards angry facial expressions.	d = 0.49
								-The moderate difference between AN-R and HC on disgust response towards angry facial expressions.	d = 0.68
								-Following depression score covariation, AN-R patients reported elevated disgust levels towards angry facial expressions compared to BN and HC (df =1, t =22.58, *p* = 0.013).	N/A
**Hay and Katsikitis (2014)** [46] ****** **Australia**	F adults	26	Psychiatric control (PC: 20) HC (61)	26.1 (8.3)	PC: 30.9 (10.9) HC: 26.1 (7.7)	Case-control	VAS **(disgust towards visual stimuli: food and non-food pictures shown)**	-The disgust responses to food images in the ED group were higher than ones in either control group.	d for ED vs. PC = 1.205 ED vs. HC = 1.337
**Uher et al. (2004)** [52] ****** **UK**	F adults	AN (16) BN (10)	19	AN: 26.93 (12.14) BN: 29.80 (8.80)	26.6 (8.34)	Case-control	VAS **(disgust towards visual stimuli: food vs. non-food images; aversive vs. neutral)**	-ED patients reported a higher level of disgust towards food stimuli than HC.	d = 1.66
**Joos et al. (2011a)** [49] ****** **Germany**	F adults	AN-R (11)	11	25.0 (5.0)	26.0 (5.2)	Case-control	VAS **(disgust towards visual stimuli: food or non-food pictures)**	-AN-R patients reported a higher level of disgust towards food photographs than HC.	d = 1.975
								- AN-R patients’ disgust levels increased when viewing high-calorie food photographs.	N/A
**Joos et al. (2011b)** [50] ****** **Germany**	F adults	Medication-free BN (13)	13	25.2 (5.1)	27.0 (6.0)	Case-control	VAS **(disgust towards visual stimuli: food or non-food pictures)**	- BN patients reported a higher level of disgust towards food photographs than HC.	d = 0.67
**Horndasch et al. (2012)** [47] **Germany**	F adolescents	AN (13)	Typically developing girls (18)	15.7 (1.8)	16.6 (1.8)	Case-control	VAS **(disgust towards visual stimuli: underweight, normal, and overweight female body pictures)**	-Both groups reported higher levels of disgust towards underweight and overweight body pictures compared to normal ones.	d for Normal vs. under-weight in AN = 1.25 Normal vs. over-weight in AN = 1.55 Normal vs. under-weight in HC = 1.51 Normal vs. over-weight in AN = 1.69
**Schienle et al. (2004)** [57] **Germany**	F adults	BN (11)	12	25.4 (9.0)	26.3 (6.4)	Case-control	VAS **(disgust ratings for visual stimuli: disgust- vs. fear-inducing vs. neutral pictures)**	-BN patients found disgust-inducing pictures as highly repulsive as fear-inducing ones.	d = 1.35
**Gagnon et al. (2018)** [56] **Canada**	F adults	AN-R (5) AN-BP (5) BN (13)	23	General: 30.35 (11.31)	25.91 (5.86)	Case-control	Temporal Bisection Task as a time perception task during **visual stimuli: disgusting vs. joyful vs. neutral food pictures**	-AN patients tended to perceive the duration of disgusting food pictures longer than those with BN.	d = 0.99
								-AN patients tended to perceive the duration of disgusting food pictures longer than neutral ones.	d = 0.53
**Foroughi et al. (2020)** [45] ****** **Australia**	F adults	AN (61) AAN (26) BN (39) BED (13)	41	AN: 16.47 (2.08) AAN: M = 18.92 (7.53) BN: 25.20 (10.26) BED: 31.38 (15.13)	21.02 (8.02)	Case-control	VAS **(disgust towards visual stimuli: disgust-eliciting food and non-food images shown by PowerPoint slides)**	-Disgust ratings towards food images were higher in individuals with ED than HC but did not differ between ED groups.	d for HC vs. EDs = 1.52 AN vs. AAN vs. BN vs. BED = 0.53
		AN and AAN (180) -pre-treatment -post weight-gain	N/A	N/A		Longitudinal		-Following weight gain, the disgust of AN and AAN patients towards food images declined, but it remained higher than HC.	d = −0.78

**Abbreviations:** NI: No Information, N/A: Not Applicable, F: Female, BN: Bulimia Nervosa, AN: Anorexia Nervosa, AN-R: Anorexia Nervosa-Restrictive Type, AN-BP: Anorexia Nervosa: Binge–Purge Type, BED: Binge Eating Disorder, PC: Psychiatric Control, ED: Eating Disorder, EDs: Eating Disorders, HC: Healthy Control, Age M: Age Mean, SD: Standard Deviation, VAS: Visual Analogue Scale. **Footnotes:** * The study provided outcomes under the category of aversive emotion involving disgust and fear due to the high correlation between ratings of both emotions with each visual stimulus. ** The study was included in our meta-analyses.

**Table 3 nutrients-14-01728-t003:** A summary of studies using experimental tasks to measure cognitive–emotional aspects of disgust.

Author (Year) Country	Gender (*n*)	Sample Size		Age M (SD)		Study Design	Method	Main Findings	Effect Size (Cohen’s d) of Main Findings
		Clinical	Control	Clinical	Control				
**Pollatos et al. (2008)** [65] **Germany**	F adults	AN-R (12)	12	23.86 (4.25)	22.39 (4.78)	Case-control	**Emotional Face Recognition Task** (six emotional faces from KDEF by Lundqvist et al. 1998) [64]	-AN patients had lower recognition for disgusted faces compared to HC.	N/A *
**Lule et al. (2014)** [66] **Switzerland**	F adolescents	AN (15)	15	16.2 (1.26)	16.5 (1.09)	Case-control	**FEEL Test** (facial emotion recognition from JACFEE by Matsumoto and Ekman, 1988) [61]	-AN group tended to recognize disgust with less accuracy than HC (F = 3.39 *p* = 0.08), but this moderate difference disappeared with the correction of the depression score.	d = 0.70
								-Negative correlation between disgust recognition ability and the psychological characteristics “perfectionism” and “trait anxiety”.	d for Perfectionism = −1.09 Trait anxiety = −0.72
**Wyssen et al. (2019)** [67] **Switzerland and Germany**	F adults	AN (61) BN (58) A mixed group in which patients with depression (36) Anxiety (23) were found	130	AN: 22.87 (4.57) BN: 23.16 (3.96) Mixed: 25.92 (4.79)	21.53 (2.18)	Case-control	**A Computerized Emotion Recognition Task** (facial expressions of disgust from the series of Ekman and Friesen 1971) [59]	-BN group had more difficulty in recognizing expressions of disgust than HC.	d for BN vs. HC = 0.35 BN vs. AN = 0.40
								-Mixed group needed more information to accurately recognise disgust than HC and AN.	d for Mixed group vs. HC = 0.49 Mixed group vs. AN = 0.59
								-Each group had confusion between expressions of disgust and anger (17–22%).	N/A
**Duriez et al. (2021)** [68] **France**	F adults	AN (33)	33	25.03 (7.04)	26.27 (6.28)	Case-control	**Facial Emotion Recognition Task-Multi-Morph Technique** (disgust was one of the proto-typical emotions taken from the series of Ekman and Friesen, 1976) [60]	-Patients with AN accurately identified disgust more often than HC.	d = 0.60
								-Accuracy in recognizing disgust was predicted in the AN group (vs. control group) after controlling depression scores.	d = 0.58
								-Higher depressive scores were related to faster and more accurate disgust recognition among the AN group, which was not observed in HC.	d for accuracy = −0.85 speed = 0.72
								-No difference between disgust recognition performance and physical activity level.	d = 0.02
**Dapelo et al. (2016)** [69] **UK**	F adults	AN (35) Medicated patients with AN (21) Unmedicated patients with AN (14)	42	27.54 (8.36)	26.98 (7.55)	Case-control	**Facial Emotion Recognition Task** (disgust expression (Young et al., 2002) [62] depicted at different ambiguous proportions: 50%, 70%, and 90%)	-AN group manifested less accurate recognition of disgust depicted at the proportion of 90%.	Disgust recognition at 90% d = −0.85 70% d = −0.47 50% d = −0.15
								-No difference between AN and HC on emotion recognition accuracy at the proportion of 70% and 50%.	N/A
								-The minimum difference between AN and HC on response bias towards emotions shown at the proportions of 90%, 70% and 50%.	Disgust response bias at 90% d = 0.21 70% d = 0.16 50% d = 0.01
								-The minimum correlation between disgust recognition accuracy and medication situation among participants with AN.	d = 0.10
**Dapelo et al. (2017)** [70] **UK**	F adults	BN (26) AN (35)	42	27.54 (8.36)	26.98 (7.55)	Case-control	**Facial Emotion Recognition Task** (disgust expression (Young et al., 2002) [62] depicted at different ambiguous proportions: 50%, 70%, and 90%)	at the proportion of 90% -ED group showed lower disgust recognition accuracy than HC.	d for AN vs. HC at 90% = −0.93 BN vs. HC at 90% = −0.54
								at the proportion of 90% -The moderate difference among AN and BN on disgust recognition accuracy.	AN vs. BN at 90% = −0.51
								at the proportion of 70% -No difference among groups on disgust recognition accuracy.	AN vs. HC at 70% = −0.47 BN vs. HC at 70% = −0.12 AN vs. BN at 70% = −0.36
								at the proportion of 50% -No difference among groups on disgust recognition accuracy.	AN vs. HC at 50% = −0.19 BN vs. HC at 50% = −0.10 AN vs. BN at 50% = −0.10
								-BN participants misinterpreted disgust depicted at a proportion of 90% as anger compared to HC.	N/A
**Jänsch et al. (2009)** [71] **UK**	F adults	AN (28) Medicated vs. Unmedicated	28	27.11 (7.51)	28.21 (7.03)	Case-control	**Facial Expression Recognition Task** (facial expression of disgust taken from the series of Ekman and Friesen, 1976) [60] -reaction time -accuracy -misclassification	-Increased level of ED symptomatology was associated with fewer misclassification of faces as disgusted among people with and without medication.	d (with medication) = −1.58 d (without medication) = −0.63
								-Those in the medicated group recognized disgust more quickly.	N/A
								- In the unmedicated group, only accuracy for disgust decreased with a higher level of ED symptoms.	d = −1.58
**Fujiwara et al. (2017)** [75] **Canada**	F adults	AN (19) BN (5)	HC with -low alexithymia (HC-LA: 25) -high alexithymia (HC-HA: 25)	23.33 (7.12)	HC-LA: 19.92 (3.80) HC-HA: 18.60 (2.04)	Case-control	**Facial Emotion Discrimination Task** (clear vs. blended/ambiguous disgust–anger facial expressions), eye tracking)	-ED group judged ambiguous disgust–anger expressions with less accuracy than HC-LA and HC-HA. -ED group spent less time looking at disgust–anger expressions than HC-LA and HC-HA.	d (accuracy) for ED vs. HC-LA = −0.86 ED vs. HC-HA = −0.67 d (time) for ED vs. HC-LA = 0.74 ED vs. HC-HA = 0.64
								-Accuracy in judging ambiguous disgust–anger expressions was less than that of clear expressions among all participants (η^2^ _partial_= 0.24).	N/A
								-In ED group only, difficulty judging ambiguous disgust–anger faces was predicted by less visual attention to the faces (β = 0.88, t = 3.44, *p* = 0.004) and a lesser tendency to gaze at the faces’ eye (β = 0.38, t = 1.84, *p* = 0.09).	N/A
**Hildebrandt et al. (2018)** [74] **US**	F adolescents	AN (16)	N/A	16.0 (1.4)	N/A	Case-control	**Voluntary Facial Expression Task:** Simultaneous fMRI-EMG data were collected for yuck/disgust vs. happy facial expressions.	-EMG measuring the patterns of voluntary muscle activation (Levator labii that contributes to facial expression and movement of the mouth and upper lip) can be used to distinguish disgust from happiness.	N/A
								-Levator labii was more active in response to disgusted faces than Zygomaticus (Mdiff = 0.294, SE = 0.022, 95% CI = 0.251, 0.337, t = 13.39, *p* < 0.0001) and Corrugator (Mdiff = 0.091, SE = 0.022, 95% CI = 0.134, 0.048).	N/A
**Dapelo et al. (2016)** [76] **UK**	F adults	AN-R (17) AN-BP (19) BN (25)	42	AN: 27.50 (8.24) BN: 26.32 (6.64)	26.98 (7.55)	Case-control	**Pose Expression Task and Imitated Expression Task** (facial expressions of disgust from the series of Ekman and Friesen, 1976) [60]	**Non-disgust-specific findings** -Those with AN and BN were less accurate at posing facial expressions of emotions.	N/A *
								-Those with AN had lower performance than HC at imitating facial expressions, whereas BN participants did not differ from those with AN and HC.	N/A *
**Kim et al. (2014)** [72] **South Korea**	F adults	AN (31)	33	23.10 (9.35)	22.18 (2.14)	Cross-over RCT	**Visual Probe Detection Task** (disgust expression photos from KOFEE facial expression photos from the series of Park et al. 2011) [63]	- Attentional bias to the disgust stimuli was observed in AN and HC under the placebo condition.	d = 0.60
								-Oxytocin had a small effect on attentional bias in the AN group.	d = 0.42
**Hildebrandt et al. (2016)** [73] **US**	F adolescents	AN-R (21) AN-BP (11)	20	16.68 (3.14)	17.91 (2.45)	Case-control	**Emotional Go/No-Go Task** (disgusted facial expressions taken from the MacBrain face stimulus set) [58]	-Patients with AN committed more commission errors for disgust stimuli than HC.	d = 0.74
								-For patients with AN, lower testosterone predicted greater behavioural disinhibition for disgusted faces (β = −0.67, 95% CI [−1.22, −0.12]).	N/A
**Gagnon et al. (2018)** [56] **Canada**		AN-R (5) AN-BP (5) BN (13)	23	30.35 (11.31)	25.91 (5.86)	Case-control	**Temporal Bisection Task** as a time perception task during visual stimuli: disgusting vs. joyful vs. neutral food pictures	-AN patients tended to perceive the duration of disgusting food pictures longer than those with BN.	d = 0.99
								-AN patients tended to perceive the duration of disgusting food pictures longer than neutral ones.	d = 0.53

**Abbreviations:** N/A: Not Applicable, F: Female, KDEF: The Karolinska Directed Emotional Faces, JACFEE: Japanese and Caucasian Facial Expression of Emotions, BN: Bulimia Nervosa, AN: Anorexia Nervosa, AN-R: Anorexia Nervosa-Restrictive Type, AN-BP: Anorexia Nervosa: Binge–Purge Type, BED: ED: Eating Disorder, EDs: Eating Disorders, HC: Healthy Control, HC-LA: Healthy Control-Low Alexithymia, HC-HA: Healthy Control-High Alexithymia, Age M: Age Mean, SD: Standard Deviation, fMRI-EMG: Combination of Functional Magnetic Resonance Imaging with Electromyography. **Footnotes:** * Study did not report disgust outcome independently, so effect sizes were not calculated.

**Table 4 nutrients-14-01728-t004:** A summary of brain imaging and neurophysiological studies of disgust.

Author (Year) Country	Gender (*n*)	Sample Size		Age M (SD)		Study Design	Method	Main Findings	Effect Size (Cohen’s d) of Main Findings
		Clinical	Control	Clinical	Control				
**Uher et al. (2004)** [52] **UK**	F adults	AN (16) BN (10)	19	AN: 26.93 (12.14) BN: 29.80 (8.80)	26.6 8 (8.34)	Case-control	**fMRI**	**Non-disgust-specific findings:**	
								-Greater activation in the left medial orbito-frontal and anterior cingulate cortices and less activation in the lateral prefrontal cortex, inferior parietal lobule, and cerebellum in response to food images among patients with EDs compared to HC.	N/A *
								-BN patients had less activation in the lateral and apical prefrontal cortex in response to food images than HC.	N/A *
**Uher et al. (2005)** [53] **UK**	F adults	BN (9) AN (13)	18	BN: 29.6 (9.3) AN: 25.4 (10.2)	26.6 (8.6)	Case-control	**fMRI** visual stimuli: underweight, normal, and overweight female bodies in swimming costumes	- Higher aversion scores were reported in response to all body shape categories associated with greater activation in the right medial apical prefrontal cortex among ED patients.	N/A
**Schienle et al. (2004)** [57] **Germany**	F adults	BN (11)	12	25.4 (9.0)	26.3 (6.4)	Case-control	**fMRI** **Contrasts:** Disgust > Neutral Disgust > Fear Fear > Disgust visual stimuli: disgust- vs. fear-inducing vs. neutral pictures	BN patients had greater activation in the left amygdala and right cuneus when comparing disgust with neutral and fear conditions.	N/A
								- No significant difference was found between BN and HC for each contrast.	N/A
**Schienle et al. (2009)** [77] **Germany**	F adults	BED (17) BN-purging type (14)	Normal Weight (NW: 19) Overweight (OW: 17)	BED: 26.4 (6.4) BN: 23.1 (3.8)	NW: 22.3 (2.6) OW: 25.0 (4.7)	Case-control	**fMRI** visual stimuli: disgust-inducing pictures Defined ROIs: amygdala, insula, and lateral OFC Contrast: Disgust > Neutral	-Disgust pictures induced greater activation in the defined ROIs among each group.	N/A
								-Greater insula activation to disgust-inducing pictures in BN relative to OW	d = 1.52
								-Greater insula and lateral OFC activations to disgust-inducing pictures in NW relative to BED.	d for insula in BED vs. NW = 1.30 d for lateral OFC in BED vs. NW = 1.34
**Ashworth et al. (2011)** [78] **UK**	F adults	Medication-free patients with BN (12)	16	24.4 (4.8)	27.4 (5.4)	Case-control	**fMRI** visual stimuli: disgusted female and male faces (Matsumoto and Ekman, 1988) [61]	-No significant difference was found in insula and amygdala activation in response to disgusted faces between BN and HC.	N/A
								-BN patients had reduced activation in the praecuneus/cuneal cortex towards disgusted faces compared to HC.	N/A
**Joos et al. (2011a)** [49] **Germany**	F adults	AN-R (11)	11	25.0 (5.0)	26.0 (5.2)	Case-control	**fMRI** visual stimuli: food or non-food pictures	-The disgust level in response to food pictures was negatively associated with the right amygdala signal.	d = −3.10
**Hildebrandt et al. (2018)** [74] **US**	F adolescents	AN (17)	N/A	16.0 (1.4)	N/A	Case-control	**Facial Expression Recognition Task** (facial expression of disgust) -reaction time -accuracy -misclassification	-Increased levels of ED symptomatology were associated with fewer misclassifications of faces as disgusted among people with and without medication.	d (with medication) = −1.58 d (without medication) = −0.63
								**-Those in the medicated group recognized disgust more quickly.**	N/A
								**-In the unmedicated group, only accuracy for disgust decreased with a higher level of ED symptoms.**	d = −1.58
**Hildebrandt et al. (2015)** [79] **US**	F adults	AN-R (14)	15	15.05 (1.87)	17.64 (2.71)	Case-control	**Facial EMG recording** during an experimental task (reversal learning paradigm) using theoretically aversive ED stimuli (chocolate candies)	-AN-R group had a distinct spike in levator labii activation (as a disgust marker) to food-cue during the acquisition phase compared to HC.	d = 1.36
								-The number of levator labii spikes predicted impaired extinction in reversal for AN-R group only.	d = 0.93
**Soussignan et al. (2010)** [80] **France**	F adolescents	AN (16)	25	26.68 (7.30)	24.6 (6.03)	Case-control	**EMG** during visual stimuli: palatable food pictures just after subliminal exposure to facial expressions (disgust/fear)	-Subliminal disgust expressions did not prime corrugator muscle reactivity to food stimuli in fasting AN patients, but subliminal fear expressions did.	d for disgust = N/A d for fear = 0.54
**Horndasch et al. (2012)** [47] **Germany**	F adolescents	AN (13)	15.7 (1.8)	15.7 (1.8)	16.6 (1.8)	Case-control	**EEG recording** visual stimuli: underweight, normal, and overweight female body pictures	-The highest earlier (16.4 ± 10.2 μV) and late (12.0 ± 6.4 μV) Late Positive Potential (LPP) amplitudes were found for underweight body pictures in the AN group.	N/A
**Schienle et al. (2017)** [31] **Germany**	F adults	Patients with binge-eating symptoms (36)	33	No age-specific information		Experimental	**EEG recording** during both visual stimuli (food pictures) and gustatory fluid stimuli, including water (neutral tastant) and wormwood (bitter/aversive tastant)	-Atypical/Enhanced Late Positive Potentials (LPP) towards visual food images during tasting wormwood among people with binge-eating symptoms (*p* = 0.04).	N/A
**Pollatos et al. (2008)** [65] **Germany**	F adults	AN-R (12)	12	23.86 (4.25)	22.39 (4.78)	Case-control	**EEG recording** visual stimuli: emotional faces	-In the AN group, EEG recording showed increased higher N200 amplitudes to all face categories (η^2^ = 0.25) and lower P300 amplitudes in response to unpleasant emotional faces (η^2^ = 0.37), different from HC.	N/A

**Abbreviations:** N/A: Not Applicable, F: Female, BN: Bulimia Nervosa, AN: Anorexia Nervosa, AN-R: Anorexia Nervosa-Restrictive Type, AN-BP: Anorexia Nervosa: Binge–Purge Type, EDNOS: Eating Disorder Not Otherwise Specified, BED: Binge Eating Disorder, ED: Eating Disorder, EDs: Eating Disorders, HC: Healthy Control, OW: Overweight, NW: Normal weight, Age M: Age Mean, SD: Standard Deviation, fMRI: Functional Magnetic Resonance Imaging, ROIs: Region of Interest Analysis, EEG: Electroencephalogram, EMG: Electromyography. **Footnotes:** * Study did not report disgust outcome independently, so effect sizes were not calculated.

**Table 5 nutrients-14-01728-t005:** A Summary of qualitative studies in disgust.

Author (Year) Country	Gender (*n*)	Sample Size		Age M		Study Design	Method	Main Findings	Effect Size (Cohen’s d) of Main Findings
		Clinical	Control	Clinical	Control				
**Brooks et al. (1998)** [82] **Australia**	F (10) M (1) adults	BN (11)	N/A	Range: 19–53	N/A	Qualitative	Semi-structured interviews (Discourse Analytic Approach)	-Participants with BN reported disgust feelings towards themselves and their diagnosis	N/A
**Espeset et al. (2012)** [83] **Australia**	F adults	AN (14)	N/A	29.1	N/A	Qualitative	Semi-structured interviews (Grounded Theory Methodology)	Possible triggers of experiencing disgust:	N/A
								-Social situations (i.e., feeling sensitive towards criticism or negative feedback from others).	
								-Feelings of being full or satiated and eating food	
								-Touch—The association between disgust and body dissatisfaction. -The coping strategy developed for disgust feelings was “avoidance” (of food and body awareness)	
**Fox (2009)** [54] **UK**	F adults	AN-R (5) AN-BP (6)	N/A	Range: 19–51	N/A	Qualitative	Semi-structured interviews (Grounded Theory Methodology)	-Being bullied can trigger feelings of disgust towards the body.	N/A
								-Disgust and anger were linked	
**McNamara et al. (2008)** [81] **Norway**	F adults	BN with (3) and without (2) purge AN-R (1) AN-BP (1) EDNOS (3)	N/A	29.1	N/A	Qualitative	Semi-structured interviews with PowerPoint slides involving the images of high- and low-caloric foods (Theoretical Framework Approach)	-The image of “Peking duck” was related to disgust feelings due to thoughts of lack of control over this food. -The image of “block of chocolate” was related to disgust feelings because of negative autobiographical memories.	N/A

Abbreviations: M: Mean, N/A: Not Applicable, F: Female, BN: Bulimia Nervosa, AN: Anorexia Nervosa, AN-R: Anorexia Nervosa-Restrictive Type, AN-BP: Anorexia Nervosa: Binge–Purge Type, EDNOS: Eating Disorder Not Otherwise Specified, Age M: Age Mean.

**Figure 3 nutrients-14-01728-f003:**
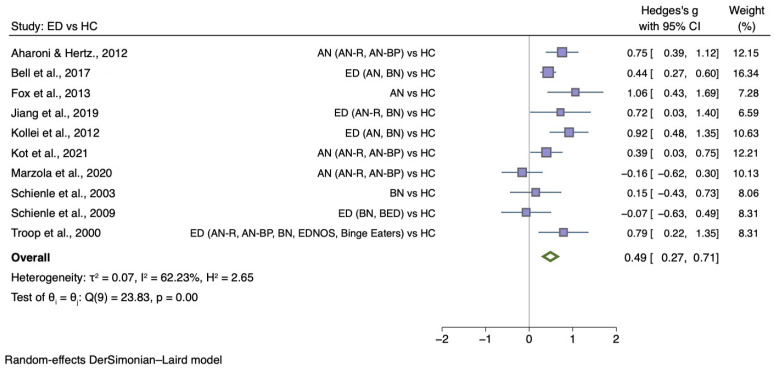
Forest plots of Hedges’ g in generic disgust sensitivity between ED participants and HCs from *n* = 10 studies; CI: confidence interval. Refs. [27,30,32,33,34,35,37,38,54,76].

The sub-group meta-analyses found that the generic disgust sensitivity score was significantly different in individuals with AN versus HC (g = 0.60; 95% CI 0.34, 0.87; *p* < 0.001) and those with BN versus HC (g = 0.50; 95% CI 0.20, 0.80; *p* = 0.004). See Figure 4 for AN meta-analysis and Figure 5 for BN meta-analysis. 

#### 5.3.3. Self-Disgust 

A total of four comparable case-control studies investigated overall self-disgust in 1196 female participants, of which 579 had an ED with a mean age ranging from 25.36 to 32.16. Two studies presented sub-type (AN and BN; [33]) or sub-scale (personal and behavioural self-disgust; [44]) values separately, and therefore, statistical data (means, standard deviations, and sample sizes) were combined for the analysis.

Figure 6 shows the meta-analysis of four studies comparing self-disgust between individuals with ED diagnosis with HC (g = 1.90; 95% CI 1.51, 2.29; *p* < 0.001). 

### 5.4. Sensitivity Analyses 

The Higgins I^2^ heterogeneity statistics indicated small to moderate (11.4% to 66.95%) heterogeneity, aside from a meta-analysis on self-disgust with 84.19%. 

For all meta-analyses, the Egger test for small-study effects was conducted to evaluate publication bias. The Egger’s test results indicated that it was not necessary to remove any smaller studies from the analysis (z = 0.09, *p* = 0.93) for disgust elicited by food images. The trim and fill correction for missing data provided no evidence for any missing studies from the analysis. 

Moreover, it was not necessary to remove any smaller studies from the analyses for generic disgust sensitivity (ED: z = 0.25, *p* = 0.81; AN: z = 1.51, *p* = 0.13; BN: z = −0.65, *p* = 0.51). The trim and fill correction for missing data was performed and revealed that there was no evidence for any missing studies on disgust sensitivity for the ED meta-analysis. However, one study for BN and two studies for AN were missing in the meta-analyses. The effect size was then re-calculated, which remained significant (BN: g = 0.60; 95% CI = 0.27, 0.92; AN: g = 0.51; 95% CI = 0.26, 0.76). 

The Egger’s test and funnel plot showed an asymmetric distribution across studies on self-disgust, suggesting significant publication bias. The trim and fill correction for missing data was also conducted, revealing that there were no missing studies from the analysis. See Appendix A for funnel plots.

## 6. Discussion

### 6.1. Summary of Evidence

A total of 52 studies were eligible for our systematic review. We could synthesise studies with regard to methodological approaches used to investigate disgust and self-disgust. However, the results of studies that explored the neuroscience underpinnings of disgust were difficult to synthesise because the experimental paradigms or techniques were so varied. Our random-effects meta-analyses revealed that self-disgust (*g* = 1.90) and disgust in response to food images (*g* = 1.32) were significantly elevated in people with EDs, whereas generic disgust sensitivity was smaller (*g* = 0.49). The largest number of studies measured generic disgust, and it was thus possible to do a separate analysis for people with anorexia nervosa and bulimia nervosa. The results were similar across these diagnoses. However, the number of studies that measured self-disgust and food-related disgust was small, and few included people with BED, so it remains uncertain as to whether these are transdiagnostic constructs. 

### 6.2. Strengths and Limitations 

To the best of our knowledge, this is the first systematic review and meta-analysis summarizing measures, outcomes, and the role of disgust and self-disgust in the field of eating disorders both qualitatively and quantitatively. Nevertheless, the results of this study should be interpreted in the context of some limitations. First, most studies focused almost exclusively on adult females with AN and BN, limiting generalisability. Similarly, studies included in our systematic review could not provide knowledge on disgust-related constructs across other eating disorder diagnoses such as ARFID, pica, and rumination disorders. Thus, further research is needed across males, gender-diverse groups, and ED diagnoses, for example, to examine whether food-disgust and self-disgust are transdiagnostic elements. Second, the methods of measuring disgust and self-disgust varied. For example, some studies investigated disgust using complex items: “feeling disgusted/depressed/guilty” or “feeling of guilty, disgusted, and angry”. More specific measures evaluating disgust may be needed using its distinctive features [84] to remove the possible confounding effect of other emotions. For example, studies measuring more objective measures such as the perception of disgust using facial EMG or facial recognition technology may hold promise [74,79]. Third, studies included in this systematic review mainly used visual stimuli, but auditory and olfactory stimuli might also be of use. Surprisingly, there are few studies on self-disgust despite this being a key trait. Finally, studies exploring the wider disgust-related experiences through qualitative methodology were limited. 

There are also potential limitations to the methodology of this systematic review and meta-analysis. First, we only included English articles, and search terms did not involve EDNOS and OSFED. Disgust is multi-faceted emotion, but our search terms did not specify terms which might include other facets of disgust, such as moral disgust [85] or sexual disgust [86]. For example, pudicity life events (i.e., events that had an element of sexual shame) were commonly found in the year that preceded the onset of anorexia nervosa [87]. Moral injury [88] or betrayal sensitivity [89] might be worth exploring. 

### 6.3. Clinical Implications 

These findings highlight the need to consider disgust in the psychopathology of eating disorders. Future studies might explore whether personality-based traits (e.g., perfectionism, cognitive, or moral rigidity, sensory sensitivity) predispose one to experiencing disgust towards food, eating and physical or behavioural characteristics of self, and possible differences across ED diagnoses. 

Furthermore, disgust may need to be considered in the treatment of eating disorders. This may involve addressing disgust experienced in response to food or eating specific stimuli or using training approaches to moderate self-disgust. However, interventions directed at these emotions specifically are not yet readily available. Existing psychotherapies address difficult emotions in general, and these may include disgust and self-disgust, but they do not provide specific guidance on how to address them across ED diagnoses. 

Traditional or adjunctive [90] exposure techniques that target aversive learning processes may be helpful to overcome disgust outcomes in response to food or eating. For example, future aversive expected outcomes might elicit a disgust response [6,15]. These may relate to weight gain or stigma and include negative evaluation from self and others [91]. 

In order to overcome future body/self-judgment expectancies that may relate to self-disgust, a different approach may be needed. A variety of novel approaches have been developed, including virtual reality used to expose the individual to the experience of being in a larger body [10] and imaginal exposure [8]. These exposures may be embedded in approaches that encourage developing an identity that includes more self-compassion, such as MANTRA [92] and compassion-focused therapy [93].

## 7. Conclusions

Our systematic review and meta-analysis suggested that disgust and self-disgust appear to be of importance in eating disorders, but to date, they have been poorly researched across the spectrum of eating disorder diagnoses and as possible predisposing or perpetuating factors. Understanding the current evidence for the role of disgust in EDs can direct future research and the development of effective treatments targeting this emotion.

## Figures and Tables

**Figure 1 nutrients-14-01728-f001:**
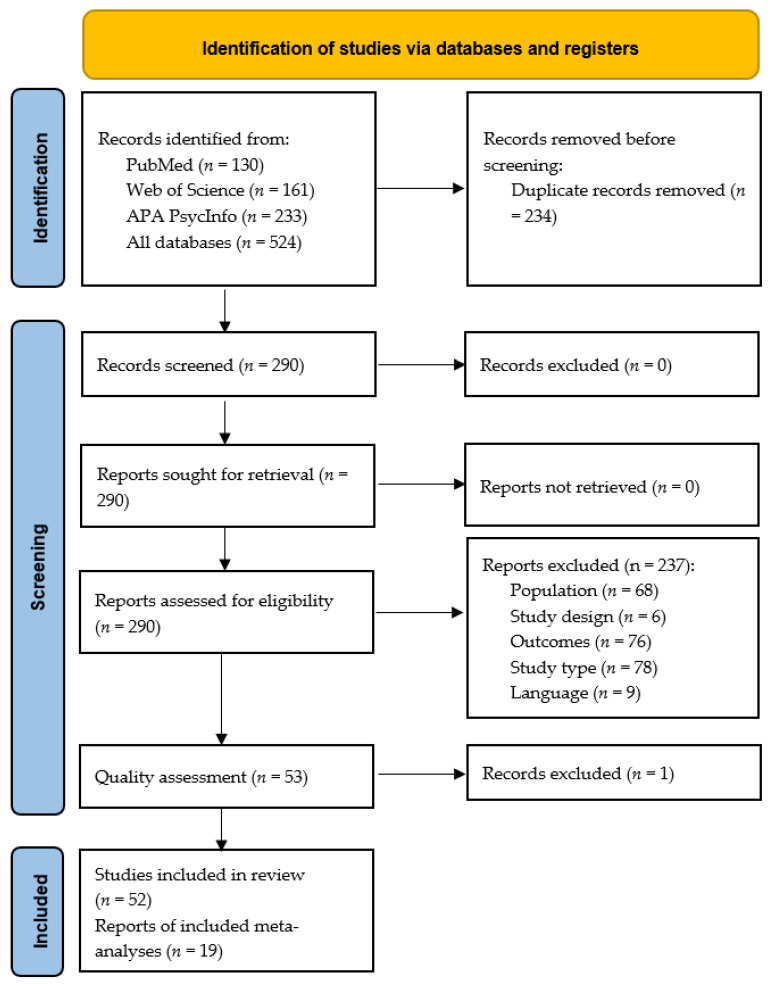
PRISMA flow diagram illustrating the process of our review, screening, and article selection processes.

**Figure 2 nutrients-14-01728-f002:**
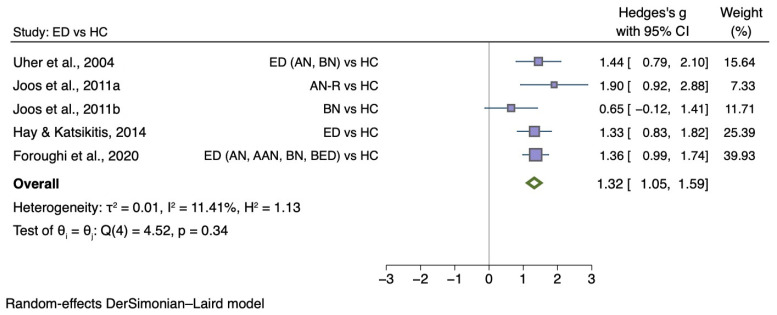
Forest plot of Hedges’ g in disgust elicited by food images between ED participants and HCs from *n* = 5 studies; CI: confidence interval. Refs. [45,46,49,50,52].

**Figure 4 nutrients-14-01728-f004:**
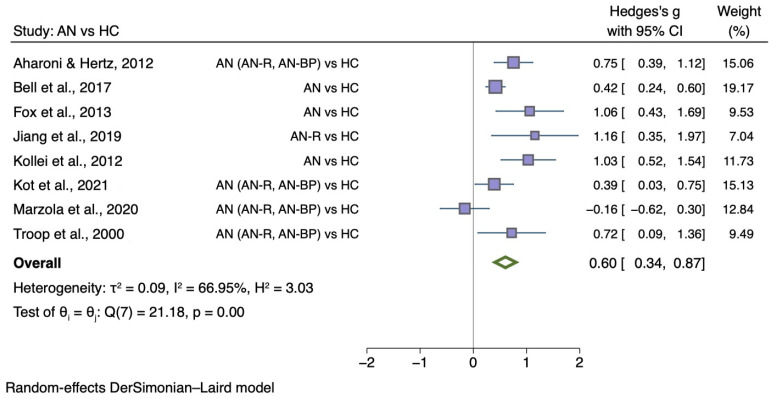
Forest plots of Hedges’ g in generic disgust sensitivity between AN participants and HCs from *n* = 8 studies; CI: confidence interval. Refs. [27,32,33,34,35,37,38,54].

**Figure 5 nutrients-14-01728-f005:**
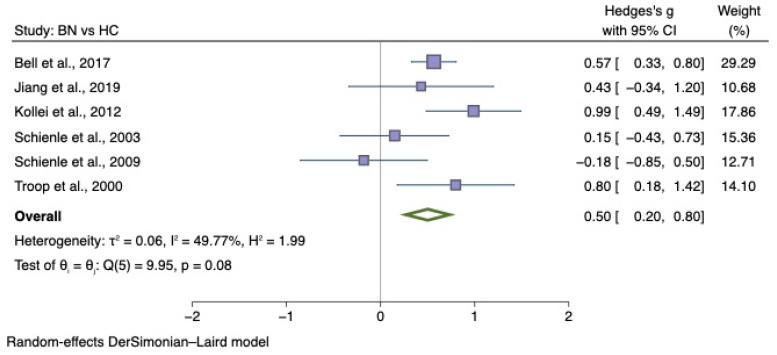
Forest plots of Hedges’ g in generic disgust sensitivity between BN participants and HCs from *n* = 6 studies; CI: confidence interval. Refs. [27,30,32,33,34,79].

**Figure 6 nutrients-14-01728-f006:**
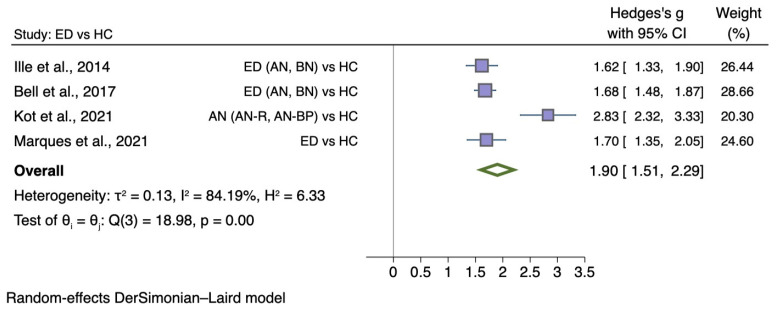
Forest plot of Hedges’ g in self-disgust between ED participants and HCs from *n* = 4 studies; CI: confidence interval. Refs. [33,38,43,44].

## Data Availability

Not applicable.

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
