# Peer review of "Disgust and Self-Disgust in Eating Disorders: A Systematic Review and Meta-Analysis"

_nutrients, 2022, doi:10.3390/nu14091728_

Round 1
Reviewer 1 Report
This paper uses a systematic review and meta-approach to examine disgust and self-disgust in those with eating disorders in comparison to healthy control groups. It is novel in taking a broad approach to concepts and measurement of disgust and also in including qualitative and quantitative studies.
The authors found evidence to support previously identified associations of high disgust sensitivity in those with eating disorders but illuminating the relative importance of the different aspects is useful. The paper is well-written, methods are sound and the results are clearly presented . The discussion appropriately contextualises these findings and raises some important topics for further research. I agree that more research is needed for BED, and also ARFID - it would be important to measures all disgust variables in those croups. They also highlight the importance of greater attention to disgust in treatment of eating disorders to reduce a potential trigger for relapse. Further examination of the link with perfectionism and rigid thinking styles would also be of interest. The authors note the female dominance in this research field but could comment on whether further research is needed for males and gender-diverse groups. I am not sure that pica and rumination disorder ought to be included in the list of eating disorders in the introduction without stating that they would be excluded or unlikely to be included in the samples reviewed. I am making assumptions here but people with those behaviours may score differently on disgust measures - an area for further research perhaps.
Minor issues
- Section 2.3- is 1806 the correct earliest date?
- Should the excluded study [25] be shown as an exclusion on the flowchart?
- Lines 403- 407. This is a little unclear. Can you please expand on this?
- There are just a few typos- p 2- final sentence repeats “were”
- Line 142 & 155 Hedge’s g or Hedges’ g? Is the (g) in parentheses needed?
- Line 159 space needed after was
- Line 231 Tables 1-5
- 238 Tables (need capital)
- Line 250 Start sentence with Two not 2
- Lines 254 & 263 – inconsistent punctuation re See Table…
- 366 remove comma
- Figure 2. Not sure that is it necessary to state that there are two Joos studies in the same year as the a b notation does that and they are clearly different samples
- The final reference Gilbert is not formatted.
- P8 F adults in the Marzola references is bolded
Author Response
This paper uses a systematic review and meta-approach to examine disgust and self-disgust in those with eating disorders in comparison to healthy control groups. It is novel in taking a broad approach to concepts and measurement of disgust and also in including qualitative and quantitative studies.
Thank you for these comments.
The authors found evidence to support previously identified associations of high disgust sensitivity in those with eating disorders but illuminating the relative importance of the different aspects is useful. The paper is well-written, methods are sound, and the results are clearly presented.
Thank you for these comments.
The discussion appropriately contextualises these findings and raises some important topics for further research.
Thank you for these comments.
I agree that more research is needed for BED, and also ARFID - it would be important to measure all disgust variables in those croups. They also highlight the importance of greater attention to disgust in treatment of eating disorders to reduce a potential trigger for relapse.
Thank you for these comments.
Further examination of the link with perfectionism and rigid thinking styles would also be of interest.
This hypothesis is of interest; we have therefore added. One possibility is of interest to investigate whether there might be a statistically significant association between self-disgust and perfectionism and other cognitive traits (e.g.,
The authors note the female dominance in this research field but could comment on whether further research is needed for males and gender-diverse groups.
Thank you for this recommendation. We have now added the sentence
“First, most studies focused almost exclusively on adult females with AN and BN, limiting generalisability. Similarly, studies included in our systematic review could not provide knowledge on disgust-related constructs across other eating disorder diagnoses such as ARFID, pica, and rumination disorders. Thus, further research is needed across males, gender-diverse groups, and ED diagnoses, for example, to examine whether food-disgust and self-disgust are transdiagnostic elements.”
I am not sure that pica and rumination disorder ought to be included in the list of eating disorders in the introduction without stating that they would be excluded or unlikely to be included in the samples reviewed. I am making assumptions here but people with those behaviours may score differently on disgust measures - an area for further research perhaps.
Thank you for this helpful comment encouraged us to delete and add information showing the lack of effectiveness from the current treatment approaches and the importance of investigating underpinning mechanisms in the introduction. Also, we mentioned generalisability of findings as limitation of our study in the discussion part (see section 6.2) by adding this sentence:
Similarly, studies included in our systematic review could not provide knowledge on disgust-related constructs across other eating disorder diagnoses such as ARFID, pica, and rumination disorders. Thus, further research is needed across males, gender-diverse groups, and ED diagnoses, for example, to examine whether food-disgust and self-disgust are transdiagnostic elements.”
Minor issues
Thank you for all below points you highlighted. Please kindly see the amendments we proceeded with based on comments.
- Section 2.3- is 1806 the correct earliest date?
In our first version of the manuscript, we wrote ‘1806’ as this is the inception date of APA PsycInfo. However, as this inception date does not apply to all the databases we used, we changed the wording. We are now writing ‘from inception.’
Should the excluded study [25] be shown as an exclusion on the flowchart?
This has been amended with the addition of a new box showing the quality assessment part. Also, we modified the flow diagram by adding two more boxes entitled “records excluded” and “reports not retrieved,” as recommended by PRISMA.
Lines 403- 407. This is a little unclear. Can you please expand on this?
We have now amended sentences. Please kindly see the latest version: “Two studies presented sub-type [33] or sub-scale [44] values separately. For our analysis, we combined statistical data (means, standard deviations and sample sizes).”
There are just a few typos- p 2- final sentence repeats “were”
We have removed the second “were” and added new punctuation “:” following the word of inclusion.
Line 142 & 155 Hedge’s g or Hedges’ g? Is the (g) in parentheses needed?
Each “Hedge’s g” written in text and figure explanation of meta-analysis has been altered to “Hedges’ g.”
Line 159 space needed after was
We added the space needed.
Line 231 Tables 1-5
“Table 1-5” has been altered to “Tables 1-5”
238 Tables (need capital)
“tables” has been altered to “Tables”
Line 250 Start sentence with Two not 2
It has been amended.
Lines 254 & 263 – inconsistent punctuation re See Table…
All inconsistent punctuation marks found in lines 254&263 have been corrected.
366 remove comma
Comma has been removed.
Figure 2. Not sure that is it necessary to state that there are two Joos studies in the same year as a b notation does that and they are clearly different samples
Thank you for your clarification because we were concerned about a possible misunderstanding of readers. This notation has been removed based on your opinion.
The final reference Gilbert is not formatted.
The final reference has been re-formatted.
P8 F adults in the Marzola references is bolded
Thank you for this point. We recognized other inconsistent bolded parts seen across tables and amended each one as you highlighted. Also, we corrected grammar mistakes in the result tables.
In addition,
We have now added one more heading from PRISMA for clarity, and we have made minor additional editing changes to improve the flow in the discussion section.
We could synthesize the findings of qualitative studies (kindly see section 5.2.2.) and highlight the limited number of qualitative studies. Also, we added methodological and data collection information regarding qualitative studies in section 5.2.1.5.
To improve the quality of meta-analysis figures, we have made minor amendments (changing default font and font sizes of specific diagnosis information) and changed the format of figures from png to jpg (high quality).
We modified Figure 1 and all result tables.
We recognised mistakes in supplementary table 1 and modified them accordingly.
We modified references following the discussion revision.
We corrected grammar, spelling mistakes or unclear sentences found in the text.
Kindly see the attached file showing our revised manuscript.

Reviewer 2 Report
In the manuscript “Disgust and Self-Disgust in Eating Disorders: A Systematic Review and Meta-Analysis” the Authors tried to synthesize involving the disgust and self-disgust in people with eating disorders. The advantage of the study is an important topic, and way of study, i.e. a systematic reviews and also meta-analysis. The systematic review of the literature revealed 52 original research articles, which is a large number. According to the Authors, disgust has been less extensively studied, therefore the Authors focus on a subject that is little recognized. Moreover, the Authors used three databases, which is proper. The review was registered in the Open Science Framework and followed PRISMA guideline.
Generally, the manuscript provides interesting information. However, I have some doubts.
The first general concern is that the Discussion section is not described in a comprehensive manner. This does not even resemble a discussion. The Authors should improve it.
The elements of a scientific discussion are:
1/ Making assertions and justifying them.
2/ Critique of claims and justifications, either by pointing out errors or by completing and indicating new points of view.
3/ Gradual clarification, ordering of thoughts.
4/ Agreeing on positions, which may result in the recognition of a certain view on the considered issue.
The same applies to the Conclusion section. How can the information obtained be further used? Recommendations for further research should be outlined, and further questions that arose during the study should be summarized.
I do not see whole letters in figure 1. I mean “Identification of studies via databases and registers”.
You have double “the” in line 241.
Could you improve the quality of all figure involving a meta-analysis?
Author Response
In the manuscript “Disgust and Self-Disgust in Eating Disorders: A Systematic Review and Meta-Analysis” the Authors tried to synthesize involving the disgust and self-disgust in people with eating disorders. The advantage of the study is an important topic, and way of study, i.e. a systematic reviews and also meta-analysis.
Thank you for these comments.
The systematic review of the literature revealed 52 original research articles, which is a large number. According to the Authors, disgust has been less extensively studied, therefore the Authors focus on a subject that is little recognized.
We are confused by this statement. Although the number of studies seems large, disgust-related constructs, particularly self-disgust, were poorly researched across the spectrum of eating disorder diagnoses. Also, studies exploring disgust-related experiences through qualitative methodology were limited. Following this statement, we synthesized findings of qualitative studies (kindly see section 5.2.2.) and highlighted the limited number of qualitative studies. Also, we added methodological and data collection information regarding qualitative studies in section 5.2.1.5.
Moreover, the Authors used three databases, which is proper. The review was registered in the Open Science Framework and followed PRISMA guideline.
Generally, the manuscript provides interesting information. However, I have some doubts.
Thank you for these comments.
The first general concern is that the Discussion section is not described in a comprehensive manner. This does not even resemble a discussion. The Authors should improve it.
The elements of a scientific discussion are:
1/ Making assertions and justifying them.
2/ Critique of claims and justifications, either by pointing out errors or by completing and indicating new points of view.
3/ Gradual clarification, ordering of thoughts.
4/ Agreeing on positions, which may result in the recognition of a certain view on the considered issue.
We are puzzled by these statements since our discussion conforms with the guidelines set out in the paper by Steward et al. (2015; kindly see the link: https://pubmed.ncbi.nlm.nih.gov/25919529/) Also, in accordance with the author guidelines of the journal Nutrients, we followed the PRISMA guideline (2020; kindly see the link: https://systematicreviewsjournal.biomedcentral.com/articles/10.1186/s13643-021-01626-4) when drafting the discussion. Therefore, the discussion summarized the main results of our systematic review and meta-analysis, discussed the limitations of the literature obtained and the limitations of our own scientific approach and explained the implications for further research. Thus, we are afraid that following this suggestion will breach the PRISMA guidelines.
However, we have now added a new heading (see the section 6.1. summary of evidence) from PRISMA for clarity and we have made minor additional editing changes to improve the flow. Kindly see the new revision of manuscript.
The same applies to the Conclusion section. How can the information obtained be further used? Recommendations for further research should be outlined, and further questions that arose during the study should be summarized.
We are also afraid that following the suggestion for the conclusion because we examined the last 1-year published systematic review and meta-analyses in the journal of Nutrients. The majority of studies include the conclusion part without the structure suggested by the reviewer. However, we have made minor additional editing changes.
I do not see whole letters in figure 1. I mean “Identification of studies via databases and registers”.
We did not understand the content of this comment; however, we amended the figure to make it clear as much as possible by proceeding with these changes:
- The removal of quality assessment from the box for the reports excluded and the addition of a separate box showing the quality assessment step.
- The addition of two more boxes (as recommended by PRISMA): “Records excluded” and “Reports not retrieved,” even though these were not applicable for our study.
- Also, we changed the title of Figure 1 as "PRISMA flow diagram illustrating the process of our review, screening, and article selection processes."
You have double “the” in line 241.
Second “the” has been removed.
Could you improve the quality of all figure involving a meta-analysis?
In our first version of manuscript, we copied “png” versions of figures into the text. Based on your suggestion, we modified meta-analysis figures by changing format of additional information (showing sample details for each study) written on the figure, and to improve their quality, we used high quality jpg format. We would be pleased to share original figures separately.
In addition,
We modified all result tables.
We recognised mistakes in supplementary table 1 and modified them accordingly.
We modified references following the discussion revision.
We corrected grammar, spelling mistakes or unclear sentences found in the text.
Kindly see the attached file showing the revised manuscript.

Round 2
Reviewer 2 Report
Thank you for your all answers.